# Interference-Free Measurement of Urinary Angiotensin-Converting Enzyme (ACE) Activity: Diagnostic and Therapeutic Monitoring Implications

**DOI:** 10.3390/biomedicines13102528

**Published:** 2025-10-16

**Authors:** Attila Ádám Szabó, Enikő Edit Enyedi, Tamás Bence Pintér, Ivetta Siket Mányiné, Csongor Váradi, Emese Bányai, Attila Tóth, Zoltán Papp, Miklós Fagyas

**Affiliations:** 1Division of Clinical Physiology, Department of Cardiology, Faculty of Medicine, University of Debrecen, 4032 Debrecen, Hungary; szabo.attila@med.unideb.hu (A.Á.S.); pinter.tamas@med.unideb.hu (T.B.P.);; 2Kálmán Laki Doctoral School of Biomedical and Clinical Sciences, University of Debrecen, 4032 Debrecen, Hungary; 3Department of Surgery, Faculty of Medicine, University of Debrecen, 4032 Debrecen, Hungary; 4Department of Emergency Medicine, Faculty of Medicine, University of Debrecen, 4032 Debrecen, Hungary

**Keywords:** ACE inhibitors, therapy adherence, urinary biomarker

## Abstract

**Background/Objectives**: Urinary angiotensin-converting enzyme (uACE) activity has long been regarded as a promising biomarker for kidney and cardiovascular diseases; however, its clinical applicability has been limited by the presence of endogenous urinary inhibitors and technically demanding assay protocols. We aimed to establish a fast and reproducible method for measuring uACE activity to identify the inhibitory compounds responsible for previous assay failures and to define practical preanalytical conditions suitable for routine laboratory implementation. **Methods**: A fluorescence-based kinetic assay was optimized for urine samples. Endogenous inhibitors were isolated by membrane filtration and chemically characterized, while the effect of sample dilution was evaluated as a simplified alternative for eliminating inhibitory interference. We assessed the stability of ACE activity under various storage conditions to support reliable measurement. **Results**: Urea (IC_50_ = 1.18 M), uric acid (IC_50_ = 3.61 × 10^−3^ M), and urobilinogen (IC_50_ = 2.98 × 10^−4^ M) were identified as the principal reversible inhibitors, jointly accounting for up to 90% suppression of uACE activity. Their inhibitory effect was effectively eliminated by a 128-fold dilution. ACE activity remained stable for 24 h at 25 °C but was completely lost after freezing. A strong positive correlation between uACE activity and creatinine concentration (r = 0.76, *p* < 0.0001) justified normalization. ACE activity-to-creatinine ratio turned out to be significantly lower in ACE inhibitor-treated patients than in untreated controls (6.49 vs. 36.69 U/mol, *p* < 0.0001). **Conclusions**: Our findings demonstrate that accurate measurement of uACE activity is feasible using a rapid dilution-based protocol. The normalized ACE activity can serve as a practical biomarker for detecting pharmacological ACE inhibition and monitoring therapy adherence in cardiovascular care and may also provide insight into renal pathophysiology such as tubular injury or local RAAS-related processes.

## 1. Introduction

Angiotensin-converting enzyme (ACE, kininase II, peptidyl-dipeptidase A, CD143, EC3.4.15.1) is a zinc metallopeptidase first described by Skeggs et al. in the 1950s [1]. ACE has been in the focus of scientific research for decades as it has been found to be a key component of the renin–angiotensin–aldosterone system (RAAS), which is a significant regulator of blood pressure and salt-water homeostasis [2,3].

ACE is highly expressed on the vascular endothelial surface of various organs [4,5], as well as on the apical surface of epithelial cells in the kidney and small intestine [6,7,8], on neuroepithelial cells [9], and in monocytes [10]. Numerous experiments underscored the crucial role of ACE in the pathogenesis of diseases [11,12,13] with high morbidity and mortality [14,15]; therefore, ACE became a potential pharmacological target [16,17,18].

Angiotensin-converting enzyme inhibitors (ACEis) have been among the most prescribed drugs in the United States and the European Union for many years [19,20,21]. They are widely used in the long-term treatment of cardiovascular diseases (CVDs), such as hypertension, heart failure, and stroke [22,23], as well as for preventing renal complications in diabetes mellitus [24]. However, despite the proven efficiency of ACEi, patients’ adherence to therapy appears to be a recurrent problem in western healthcare systems [25,26].

As shown, the endothelial localization of ACE is well understood and exploited in the clinic, although other functions of the enzyme are still the subject of research [7,27,28]. ACE is expressed on the brush border of renal proximal tubular epithelial cells and may enter urine at least in part by shedding [29,30], although additional contributions from filtered plasma-derived ACE cannot be excluded. Its putative roles in tubular bradykinin inactivation and consequential water and sodium reabsorption [31], as well as in amino acid and bicarbonate reabsorption have been implicated [32,33].

An increase in luminal ACE release was found in different diseases associated with damage to the proximal tubular epithelium [34,35]. However, the utility of urinary angiotensin-converting enzyme (uACE) level as a biomarker has been hampered by the presence of endogenous inhibitors in urine that interfere with the measurement of ACE activity [36]. Furthermore, the methods developed to eliminate these inhibitors appeared to be extremely laborious, time-consuming, and hardly reproducible [7,37].

Here, we aimed to develop a reliable and efficient method for measuring uACE activity that overcomes previous limitations caused by endogenous inhibitors. We sought to identify these interfering compounds, establish a rapid and reproducible assay using modern fluorescent substrates and explore the method’s potential utility not only for diagnostic applications but also for monitoring pharmacological effects, such as ACEi therapy.

## 2. Materials and Methods

### 2.1. Subjects and Ethical Approval

During assay development and when aiming at the identification of endogenous inhibitors, urine samples were obtained from healthy staff members of our research department (n = 5). Following this methodological phase and the identification of physiological relationships, a prospective comparative study was conducted between two groups: a control population (n = 23), consisting of healthy individuals and CV patients not receiving ACEi, and a study population (n = 33) of CV patients treated with ACEi at the University of Debrecen, Clinical Center (Debrecen, Hungary). Three individuals from the control group were excluded due to exogenous creatine intake or special high creatine diet. The final control group included 20 individuals (Table 1). Patients gave written informed consent to urine sampling, uACE activity and creatinine measurements before being included into the study. To protect patients’ personal data, unique study codes were used to ensure patients’ anonymity from the evaluators. The clinical study was approved by the Scientific and Research Ethics Committee of the University of Debrecen and the Ministry of Human Capacities. Research authorization number: 15471-6/2023/EÜIG (approved on 4 July 2023). The research was in accordance with the tenets of the Helsinki Declaration.

### 2.2. Urine Samples

First morning midstream urine samples and daytime dilute urine samples were drawn on the day of measurements using standard urine sampling techniques. Preanalytical and analytical steps were performed at the Division of Clinical Physiology, Department of Cardiology, Faculty of Medicine, University of Debrecen, Debrecen, Hungary. Samples were transported and centrifuged (10 min; 400× *g*) at 25 °C. All determinations were performed on sediment-free fresh urine samples.

### 2.3. ACE Activity Measurement with the Fluorescent Kinetic Assay

ACE activity was measured using the optimized fluorescent kinetic method previously reported by our group [38] with different sample dilutions for each experiment.

In the final dilutional protocol, the reaction mixture contained 100 mM tris(hydroxymethyl)aminomethane hydrochloride (TRIS) buffer (pH 7.0), 15 μM Abz-FRK(Dnp)P-OH substrate (synthesized by Peptide 2.0, Chantilly, VA, USA), 50 mM NaCl, 10 μM ZnCl_2_, and urine at 64-fold and 128-fold final dilutions. A control for each reaction mixtures containing 1 µM lisinopril (Cat. No: L6292-100 mg; Sigma-Aldrich, St. Louis, MO, USA) was also included. Continuous recording of fluorescent intensity change was performed for 90 min with 1-min measurement intervals at 37 °C. Data points between 20 min and 80 min were used for analysis. When activity measured at 64-fold dilution exceeded that measured at 128-fold, the 64-fold results were used to avoid over-dilution bias of dilute urine samples.

In membrane filtration experiments, endogenous inhibitors were separated from uACE using Vivaspin 20 membrane filter columns (Sartorius Stedim Biotech GmbH, Göttingen, Germany). Native urine samples (6 mL) were diluted with distilled water to a final volume of 20 mL. The columns were centrifuged at 4000× *g* for 45 min at 25 °C. Subsequently, each column was washed twice with 20 mL of distilled water, followed by centrifugation under identical conditions (4000× *g*, 45 min, 25 °C). The combined filtrates (total volume: 60 mL) were concentrated by vacuum centrifugation to the original 6 mL volume. The retentate was recovered from the membrane using 6 mL of 100 mM TRIS buffer (pH 7.0). Both native urine and membrane-purified ACE samples were analyzed at a 4-fold dilution in comparative activity measurements.

Inhibitor-free ACE pre-purified on a 30 kDa Vivaspin 20 membrane filter column was used to identify inhibitors. Potential inhibitors were dissolved according to factory instructions (Urea: Cat. No.: U6504-1KG; Sigma-Aldrich, St. Louis, MO, USA; Uric acid: Cat. No.: U2625-25G; Sigma-Aldrich, St. Louis, MO, USA; Urobilinogen: Cat. No: 651-10; Lee Biosolutions, Maryland Heights, MO, USA). In the corresponding experiments, uACE with activities ranging from 0.15 to 0.53 U/L was applied. The obtained curves were fitted using the ‘two-phase decay’ equation in GraphPad Prism software, version 9 (GraphPad Software Inc., San Diego, CA, USA).

### 2.4. Sensitivity Testing of uACE Activity Assay

The sensitivity of uACE activity measurement was tested using ACE-specific inhibitors (lisinopril, Cat. No: L6292-100 mg, Sigma-Aldrich, St. Louis, MO, USA; captopril, Cat. No: C4042, Sigma-Aldrich, St. Louis, MO, USA) and inhibitors acting on other proteases (Amastatin hydrochloride, Cat. No: SC-202051, Santa Cruz Biotechnology, Dallas, TX, USA; Apstatin, Cat. No: BML-PI145-0001, Enzo Biochem Inc., Farmingdale, NY, USA; Bestatin hydrochloride, Cat. No: B8385, Sigma-Aldrich, St. Louis, MO, USA; MLN-4760, Cat. No: 5.30616, Merck KGaA, Darmstadt, Germany; Phosphoramidon disodium salt, Cat. No: R7385, Sigma-Aldrich, St. Louis, MO, USA; Sacubitrilat, Cat. No: HY-17620, MedChemExpress, Monmouth Junction, NJ, USA; Z-prolyl-prolinal, Cat. No: BML-PI112, Enzo Biochem Inc., Farmingdale, NY, USA).

For the immunodepletion experiments, urine samples were first depleted of endogenous inhibitors using 10 kDa filtration units. The inhibitor-free urine samples were then incubated with an ACE-specific antibody (final concentration 120 ng/mL, Cat. No: 841365, component of kit DY929, R&D Systems Inc., Minneapolis, MN, USA) and Protein G Sepharose 4 Fast Flow resin (10% final packed bead volume, Cat. No: 17-0618-001, Merck KGaA, Darmstadt, Germany). The antibody incubation was performed for 16 h at room temperature, followed by a 2-h incubation with the resin at room temperature under continuous shaking. ACE-depleted and control urine samples were subsequently assayed for activity at 4-fold dilution.

### 2.5. Measurement of Creatinine Concentration

Creatinine concentration was measured using a Cobas Integra 400 plus clinical chemistry analyser (Roche Diagnostics GmbH, Mannheim, Germany) with CREJ2 Creatinine Jaffé Gen.2 cassettes (Ref.: 04810716190, Roche Diagnostics GmbH, Mannheim, Germany).

### 2.6. Statistical Analysis

Data from first five figures are presented as mean ± standard deviation (SD), where n indicates the number of independent experiments performed. For last two figures, n refers to the number of individuals in each group. Since these data were not normally distributed, results are shown as medians (central line) with interquartile ranges (25th–75th percentiles, indicated by the lower and upper whiskers), together with all individual data points. Normality of ACE activity/creatinine concentration ratios was tested using D’Agostino-Pearson normality test. Ratios of ACEi-treated patients and control individuals were compared using the Mann–Whitney U test. Group comparisons were performed using one-way ANOVA with Holm-Šídák’s multiple comparisons test or Kruskal–Wallis test with Dunn’s multiple comparisons test. *p* values < 0.05 were considered statistically significant. Statistical analyses were performed using GraphPad Prism version 9 (GraphPad Software Inc., San Diego, CA, USA).

### 2.7. Role of Generative AI and AI-Assisted Technologies in the Writing Process

During the preparation of this work, the authors used ChatGPT (version 4.0, OpenAI, San Francisco, CA, USA) to improve the clarity, structure, and readability of the manuscript, including the abstract and cover letter. After using this tool, the authors have carefully reviewed and edited the content and take full responsibility for the integrity and accuracy of the publication.

## 3. Results

### 3.1. Only Low-Molecular-Weight (<5 kDa) Endogenous ACE Inhibitors Are Present in Urine

Urine samples were filtered using membrane filter columns with different molecular-weight cut-offs (5, 10, 30, 50 kDa), ACE activity was then measured in the corresponding retentates to estimate the apparent molecular weight of endogenous ACE inhibitors. The native (unfiltered) urine sample exhibited >92% ACE inhibition compared to the retentate obtained after filtration through the 10 kDa column (ACE activities: 7.79 ± 6.27% vs. 100 ± 17.43%, respectively; *p* < 0.005). Filtration through the 5 kDa membrane led to a marked, though incomplete, increase in ACE activity (71.40 ± 4.03%). Complete separation of endogenous uACE inhibitors was achieved using the 10 kDa filter, as no further increase in ACE activity was observed with the 30 or 50 kDa filters (100.00 ± 17.43%, 102.10 ± 18.87%, and 103.10 ± 18.85%, respectively; Figure 1a).

Successful isolation of endogenous ACE inhibitors from urine was tested by re-adding the inhibitors to the retentate (Figure 1b). Removal of inhibitors using a 10 kDa pore-size filter increased uACE activity by more than tenfold, and no significant ACE activity was detected in the filtrate. When the <10 kDa filtrate was recombined with the 10 kDa retentate, ACE activity decreased significantly, approaching the level observed in native urine (10.00% ± 5.80% vs. 7.79% ± 6.27%).

Experimental results suggested the presence of endogenous ACE inhibitors in the <5 kDa and 5–10 kDa molecular-weight ranges. The potential presence of a uACE inhibitor in the 5–10 kDa range was investigated by fractionating the <10 kDa urine component into <5 kDa and 5–10 kDa subfractions using sequential membrane filtration. Native urine was first passed through a 10 kDa membrane filter, and the resulting <10 kDa filtrate was subsequently processed using a 5 kDa membrane filter column. This procedure yielded three distinct components: the 10 kDa retentate containing uACE, the 5 kDa retentate representing the 5–10 kDa subfraction, and the <5 kDa filtrate. ACE activity was measured in each component, as well as in recombined samples where each subfraction was added back to the 10 kDa retentate. Neither the <5 kDa nor the 5–10 kDa subfraction exhibited ACE activity. Recombination of the 10 kDa retentate with the <5 kDa filtrate led to a marked reduction in ACE activity, closely reproducing the inhibition observed in native urine (12.06% ± 6.62% vs. 10.84% ± 3.55%). In contrast, addition of the 5–10 kDa subfraction had no effect on ACE activity (99.62% ± 6.67% vs. 100.00% ± 4.73%). These findings do not support the presence of an ACE inhibitor in the 5–10 kDa range (Figure 1c).

### 3.2. Dilutional Reversibility and Pharmacological Inhibition of uACE Activity

Endogenous ACE inhibition in urine is reversible, as the inhibitors can be removed by filtration. To identify a simpler alternative to filtration, we investigated whether the inhibitory effect could be eliminated by diluting the urine sample, thereby reducing the local concentration of the inhibitors. Serial dilutions of native urine were prepared, and ACE activity was measured at each dilution step. The degree of urine dilution in the reaction mixture significantly influenced the measured ACE activity. A gradual increase in dilution-corrected ACE activity was observed up to a 128-fold dilution, beyond which residual endogenous inhibition was no longer detectable (Figure 2a). Further sample dilution had a negligible effect, with only a +2.34% change in ACE activity at the 256-fold dilution. uACE activity was completely inhibited by lisinopril with an IC_50_ = 2.034 × 10^−9^ M (Figure 2b), and by captopril with an IC_50_ = 4.497 nM (Figure 2c).

### 3.3. Sensitivity Testing of uACE Activity Assay

The specificity of the established assay for uACE was tested using a series of peptidase inhibitors. Amastatin hydrochloride, an inhibitor of aminopeptidase A and aminopeptidase N, had no effect on the measured activity (Figure 3a). Similarly, apstatin and bestatin hydrochloride, inhibitors of Xaa-Pro aminopeptidase (aminopeptidase P) and aminopeptidase B/N, respectively, did not alter uACE activity (Figure 3b,c). The ACE2-selective inhibitor MLN-4760 also had no effect on the assay (Figure 3d). Sacubitrilate, a clinically used neprilysin inhibitor, did not decrease the measured activity either (Figure 3e). Likewise, the prolyl endopeptidase inhibitor Z-prolyl-prolinal showed no detectable effect (Figure 3f).

In contrast, phosphoramidon disodium salt—another neprilysin inhibitor similar to sacubitrilate—reduced uACE activity by approximately 35% at 1 μM concentration (Figure 3g). A comparable inhibitory effect of phosphoramidon was also observed on serum ACE activity (Figure 3h).

Finally, immunodepletion of uACE was performed using a commercially available ACE-specific antibody. Following immunodepletion, uACE activity decreased to 5.0 ± 0.6% of the initial value, whereas the negative control containing only Protein G Sepharose resin (without antibody) retained its full activity (107.7 ± 4.2%; Figure 3i).

### 3.4. Urea, Uric Acid and Urobilinogen Act as Physiological Inhibitors of uACE

We identified urea (60.06 g/mol; Figure 4a), uric acid (168.11 g/mol; Figure 4b), and urobilinogen (592.7 g/mol; Figure 4c) as physiological inhibitors of uACE. The inhibitory effects of these small molecules were tested on ACE purified from urine. Urea accounted for 10–25% of uACE inhibition within its reference concentration range and exhibited greater potency at higher concentrations, with an IC_50_ of 1.178 M. Uric acid contributed to 30–50% inhibition within its physiologicsal range and produced near-complete inhibition at higher concentrations (IC_50_ = 3.611 × 10^−3^ M). Urobilinogen reduced ACE activity by 3–20% under normal conditions but demonstrated strong inhibitory potential in hyperurobilinogenic states (IC_50_ = 2.976 × 10^−4^ M). Under pathological conditions, any of these molecules may be sufficient to abolish uACE activity.

### 3.5. Stability of uACE Activity Under Different Storage Conditions

To evaluate the diagnostic applicability of uACE activity measurement, we assessed the analyte’s stability at different storage temperatures. At 25 °C, ACE activity remained stable for more than one day without any preservative (Figure 5a). When stored at 4 °C (refrigerator temperature), no statistically significant decrease in ACE activity was observed for at least 12 h; however, a reduction of approximately 20% was detected after one day (Figure 5b). Surprisingly, freezing resulted in a complete loss of ACE activity in the samples (Figure 5c).

### 3.6. uACE Activity Correlates Well with Urinary Creatinine Concentration

Using the dilution-based uACE activity assay, we measured ACE activity in 20 ACEi-naive urine samples. ACE activity varied by nearly an order of magnitude, ranging from 0.18 to 1.10 U/L (Figure 6a).

Visual inspection revealed noticeable differences in urine color intensity, suggesting variability in sample concentration. To account for this, urinary creatinine concentrations were measured, revealing a similarly wide range across samples (5.9 to 24.6 mmol/L). A strong positive correlation was found between uACE activity and creatinine levels (Spearman r = 0.76, *p* < 0.0001; Figure 6b), indicating that sample concentration significantly influences the measured ACE activity. Therefore, normalization to urinary creatinine concentration is recommended to correct for dilutional effects and enable more accurate comparisons between samples.

### 3.7. ACE Inhibitor Treatment Reduces uACE Activity

Serum ACE activity can be markedly suppressed by ACEi therapy. To assess the effect of ACEi treatment on uACE activity, we analyzed samples from 33 patients receiving ACEi therapy. The uACE activity-to-creatinine concentration ratio was significantly lower in ACEi-treated patients (n = 33) than in untreated controls (n = 20) (6.492 U/mol [1.327–12.700] vs. 36.690 U/mol [29.200–44.300], *p* < 0.0001; Figure 7). These findings suggest that this ratio may serve as a reliable indicator of ACEi intake and could help identify patients with poor or limited therapy adherence.

## 4. Discussion

Urine is a readily accessible biological specimen that can be collected non-invasively without specialized training or equipment, in contrast to venous blood sampling. As demonstrated by the widespread use of urine dipsticks and clinical urinalysis, urine serves as a valuable matrix for diagnosing diverse conditions and monitoring therapeutic efficacy.

Building on this, we aimed to establish a reliable and reproducible method for measuring uACE activity that is unaffected by endogenous inhibitors. Historically, uACE measurement has not been feasible with simple and widely accessible techniques. To overcome this limitation, we developed a dilution-based assay employing a modern fluorescent substrate, enabling precise, quantitative detection of uACE activity. Given that most ACEi drugs are excreted mainly via the kidneys [22,39], we also investigated whether this method could reflect pharmacological ACE inhibition, positioning urine as a practical matrix for therapeutic monitoring.

Although the presence of endogenous inhibitors of uACE activity has been recognized [36], their specific identities had not been defined. In this study, we confirmed that uACE is almost completely inhibited under physiological conditions and suggested urea, uric acid and urobilinogen as the principal endogenous inhibitors of uACE.

Urea, the end-product of amino acid metabolism, is typically present at high concentrations in urine. With the increasing prevalence of high-protein, ketogenic, or paleo diets, elevated urinary urea levels should be considered during clinical assessment [40]. Uric acid, a breakdown product of purine nucleotides, is often elevated due to modern dietary habits or medication use, both of which can cause hyperuricosuria, further enhancing its inhibitory effect [41]. Urobilinogen, a byproduct of bilirubin degradation, should be taken into account in patients with liver disease or haemolytic anemia, as its inhibitory potential becomes particularly significant under these pathological conditions [42]. Together, these three low-molecular-weight compounds can account for up to 90% of physiological uACE inhibition, with an even greater impact likely under pathological conditions. While urea is an important source of assay interference, its high physiological urinary concentrations may also suppress uACE activity in vivo. This effect could be particularly relevant in elderly individuals with increased urea excretion, potentially acting as a homeostatic mechanism to reduce bradykinin degradation and thereby promote natriuresis and water excretion.

Importantly, all three molecules act as reversible inhibitors, and their suppressive effect on ACE activity can be neutralized by sample dilution. This phenomenon is analogous to the reversible inhibition of serum ACE by albumin in blood [43,44]. Our findings demonstrate that uACE activity is no longer inhibited by endogenous compounds at a 128-fold dilution, effectively representing an uninhibited state. This insight enabled the development of a straightforward fluorescence-based assay that eliminates the need for laborious and costly membrane filtration steps. Including 1 μM lisinopril in control reactions ensures specificity by completely inhibiting ACE activity. The method presented here can be readily implemented in any clinical laboratory already equipped for ACE activity assays, without the need for additional instrumentation. Furthermore, our sensitivity experiments confirmed that the assay specifically reflects uACE activity: no inhibition was observed with various peptidase inhibitors, whereas phosphoramidon reduced both urinary and serum ACE activity to a comparable extent, indicating a direct inhibitory effect. In addition, immunodepletion of uACE with a specific anti-ACE antibody almost completely abolished measurable activity, further confirming the ACE-dependence and selectivity of the method.

This simplified approach opens the possibility of routinely using uACE activity as a biomarker, particularly in contexts such as kidney injury [34,35], and provides potential for non-invasive therapeutic drug monitoring. A key consideration when applying this method is that once the sample is diluted to the 128-fold threshold, further dilution does not result in increased ACE activity, confirming the complete removal of inhibitory effects. However, if the original urine sample is already highly dilute (e.g., daytime samples, creatinine concentration < 2.5 mmol/L), the optimal measurement point may occur at a lower dilution, i.e., at the 64-fold level. In such cases, low or undetectable ACE activity at 128-fold dilution should prompt repeat measurement at 64-fold dilution to avoid underestimation caused by over-dilution.

To demonstrate the assay’s clinical applicability in routine settings, we evaluated the stability of uACE activity under various storage conditions. Our results showed that ACE activity remains stable at room temperature and at 4 °C for a period exceeding typical laboratory working hours, although a ~20% decline was observed after 24 h at 4 °C. Notably, unlike serum samples—which can typically be frozen and thawed without loss of enzymatic activity—uACE is highly sensitive to freezing. Loss of ACE activity occurred after a single freeze–thaw cycle, indicating that long-term storage is not feasible. This marked freeze–thaw instability may reflect structural differences from serum ACE, particularly altered glycosylation/sialylation patterns close to the N-terminal active site, as previously reported by our group [7]. In addition, the very low protein concentration of urine likely contributes to this instability, as demonstrated by the stabilizing effect of added albumin at concentrations approximating those found in serum (Appendix A). These observations highlight the importance of standardized pre-analytical conditions to ensure reliable uACE activity measurement. To preserve enzymatic integrity, urine samples should be kept at 25 °C and preferably analyzed on the day of collection.

uACE activity correlated strongly with urinary creatinine concentration. Simultaneous measurement of creatinine and normalization of ACE activity to creatinine levels effectively compensate for variations in urine concentration, allowing for more accurate and reliable comparisons across samples. This strategy aligns with established practices in urinary biomarker analysis, such as the use of the albumin-to-creatinine ratio, where normalization to creatinine accounts for dilutional effects [45]. However, creatinine-based normalization has important limitations, particularly in the context of chronic kidney disease (CKD) and tubular epithelial cell injury. In CKD, decreased glomerular filtration rate (GFR) is typically accompanied by lower urinary creatinine concentrations, partly due to impaired filtration, reduced muscle mass, and compromised tubular function [46]. Similarly, both acute or chronic tubular injury may also reduce urinary creatinine levels through impaired secretion and diminished concentrating capacity. These pathological changes can compromise the reliability of creatinine-based normalization, potentially leading to overestimation of analyte excretion rates in dilute samples [47]. Accordingly, in advanced CKD and/or significant tubular dysfunction, uACE activity values normalized to urinary creatinine should be interpreted with caution. Dedicated clinical studies are warranted to further evaluate and address the limitations of creatinine-based normalization in these contexts.

ACE insertion/deletion (ACE I/D) polymorphism is known to account for approximately 60% of the inter-individual variability in circulating ACE activity under physiological conditions [38]. Previous studies have demonstrated that this polymorphism also affects ACE protein expression within the kidney [48]; however, its impact on uACE activity has not been investigated in detail. In our cohort of healthy individuals, we observed substantial variability in uACE activity (ranging from 0.15 to 1.10 U/L), which may partly reflect differences in I/D genotype. Consequently, when defining reference ranges for uACE activity, it may be advisable to consider the patient’s I/D genotype, similar to the approach used when interpreting serum ACE activity levels [38].

Reducing ACE activity is a central therapeutic goal in the management of CVD, making ACEi among the most widely prescribed drug classes worldwide [19,20,21]. However, optimizing therapeutic strategies and dosing remains a clinical challenge, often complicated by suboptimal patient adherence [25,26]. While liquid chromatography–tandem mass spectrometry (LC-MS/MS) enables accurate quantification of ACE inhibitors in plasma and urine [49,50], these methods are costly, require invasive sampling, and are not readily available in routine clinical practice. Moreover, the diversity of ACEi compounds on the market necessitates drug-specific analytical protocols, each with dedicated calibrators and controls, further limiting their feasibility for widespread clinical use. In practice, self-reported adherence remains the primary, though often unreliable, source of information on ACEi intake. Assessment of ACEi therapy efficacy via blood pressure measurement may be confounded by comorbid factors such as white-coat hypertension, polypharmacy with multiple antihypertensives, or use of ACEi for non-blood-pressure indications (e.g., in renal disease or post-myocardial infarction). Despite minor differences in tertiary structure [7], uACE displays enzymatic characteristics—including its susceptibility to lisinopril and captopril inhibition (IC_50_ = 2.03 × 10^−9^ M, IC_50_ = 4.54 × 10^−9^ M, respectively)—that closely parallel those of circulating ACE [17]. These similarities support the use of uACE as a reliable surrogate for systemic ACE activity in therapeutic monitoring. Applying our method to a cohort of 20 untreated individuals and 33 patients on ACEi therapy, we demonstrated that the uACE activity-to-creatinine ratio is significantly reduced in treated patients. Notably, outliers within the treated group likely reflect missed doses prior to sampling, underscoring the potential of this approach to assess adherence. Although absolute ACE activity also differed between groups, we recommend the creatinine-normalized ratio for its robustness against dilutional variability. It should also be noted that certain ACEi undergo partial hepatic elimination (e.g., fosinopril, trandolapril); and in such cases, uACE activity–based assessment of therapy effectiveness may require particular caution, especially in patients with renal insufficiency where hepatic clearance may be enhanced.

The development of this simple and reproducible assay has important clinical implications. It may enable routine monitoring of ACEi adherence in outpatient settings, providing clinicians with an objective tool to identify non-compliant patients and tailor interventions accordingly. This is particularly relevant in chronic CV conditions such as hypertension and heart failure, where long-term ACEi therapy is essential for optimal disease control. uACE activity, on the other hand, is subject to multiple confounders (dietary factors, age, renal function, urine dilution, endogenous inhibitors), which constrain its reliability as an adherence biomarker.

In addition, the robustness and interference-free nature of the assay may open new diagnostic perspectives beyond adherence monitoring. Because uACE activity reflects both enzymatic properties and renal excretory pathways, the method could also serve as a potential biomarker of tubular injury, renal inflammation, or local RAAS-related processes. These applications may ultimately represent the most clinically relevant implications of our findings.

This study has some limitations. First, the exact cellular origin of uACE remains unclear. While proximal tubular shedding likely contributes, plasma-derived ACE may also play a role, especially in pathological conditions. Future studies, including plasma–urine correlation and biochemical profiling, will be required to clarify this. Second, uACE activity is lost after freezing, which may necessitate further biochemical characterization and limit broader research applicability of the method. Finally, the relatively small cohort size underlines the need for larger, targeted studies to establish physiological reference ranges for uACE activity and to explore its diagnostic value across pathological conditions.

## 5. Conclusions

In summary, this study presents a novel and practical approach for measuring uACE activity, overcoming long-standing limitations posed by endogenous inhibitors. By identifying key physiological inhibitors and establishing a dilution-based assay compatible with standard laboratory practice, we provide a reproducible and interference-free method with significant clinical potential. The creatinine-normalized uACE activity not only emerges as a promising biomarker for monitoring ACE inhibitor adherence but may also serve as an indicator of tubular injury, renal inflammation, or local RAAS-related processes, thus broadening its diagnostic and therapeutic relevance.

## Figures and Tables

**Figure 1 biomedicines-13-02528-f001:**
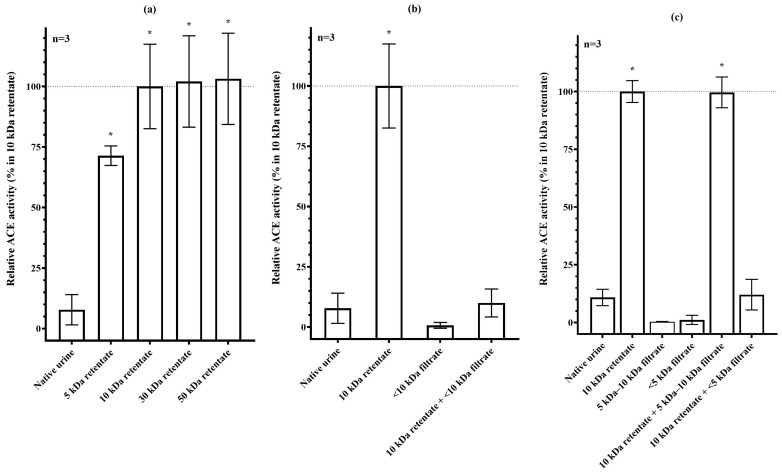
Separation of endogenous uACE inhibitors by molecular-weight filtration (**a**). uACE (>150 kDa in size) activity measured in native urine and in retentates from filtration using 5 kDa, 10 kDa, 30 kDa, and 50 kDa membrane filters. Adding back the <10 kDa filtrate inhibits uACE activity (**b**). The apparent presence of a 5–10 kDa inhibitor is only due to methodological reasons (**c**). Each bar represents the mean and standard deviation of three independent experiments. ACE activity values were compared to the activity of 10 kDa retentate. ACE activity values significantly different from uninhibited samples are indicated by asterisks (*).

**Figure 2 biomedicines-13-02528-f002:**
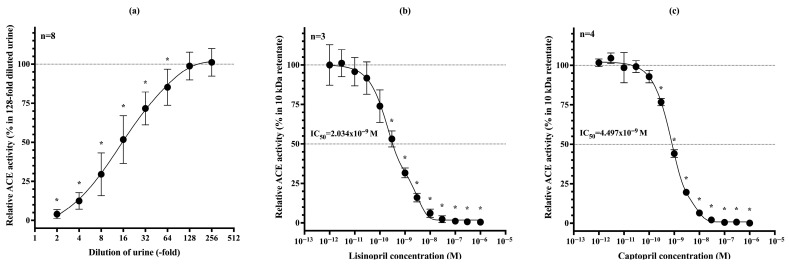
Dilution-corrected uACE activity was measured in serial dilutions (2-fold to 256-fold) (**a**). Significant increase in activity was observed up to a 128-fold dilution, beyond which no further change occurred. ACE activity values significantly different from those measured at 128-fold dilution are indicated by asterisks (*). uACE activity can be completely inhibited with ACE-inhibitor drugs, lisinopril (**b**) and captopril (**c**). The IC_50_ values for lisinopril and captopril on uACE activity are 2.034 nM and 4.497 nM, respectively. Each symbol represents the mean ± SD of at least three independent measurements.

**Figure 3 biomedicines-13-02528-f003:**
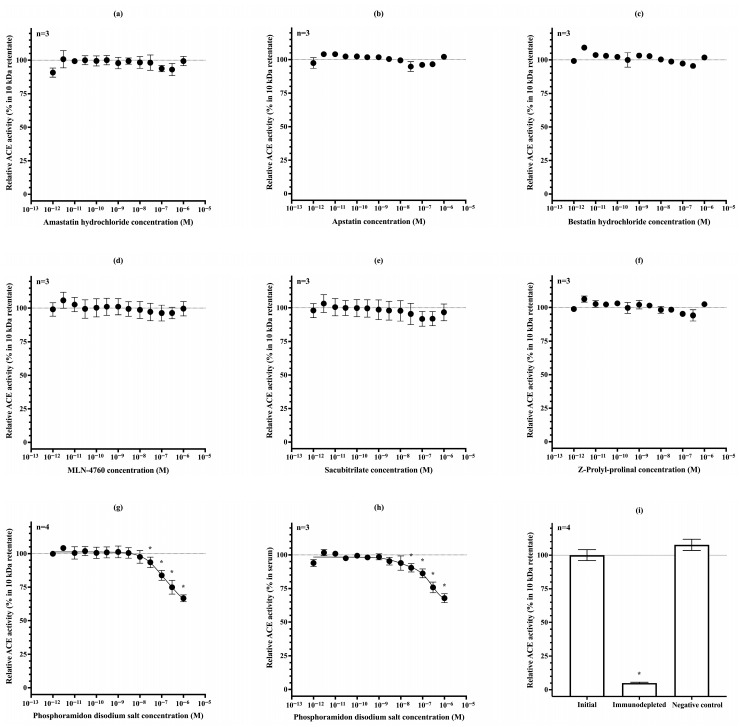
uACE activity was measured in the presence of various peptidase inhibitors: amastatin hydrochloride (aminopeptidase A and N inhibitor) (**a**), apstatin (Xaa-Pro aminopeptidase inhibitor) (**b**), bestatin hydrochloride (aminopeptidase B and N inhibitor) (**c**), MLN-4760 (ACE2 inhibitor) (**d**), sacubitrilate (neprilysin inhibitor) (**e**), Z-prolyl-prolinal (prolyl endopeptidase inhibitor) (**f**), and phosphoramidon disodium salt (neprilysin inhibitor) (**g**). The effect of phosphoramidon disodium salt on serum ACE activity is shown in subfigure (**h**). Each symbol represents the mean ± SD of at least three independent measurements, and activities are expressed as percentages of the initial inhibitor-free 10 kDa retentate. uACE activity values that are significantly different from the inhibitor-free values are indicated by asterisks (*). Subfigure (**i**) shows the residual uACE activity after immunodepletion, compared with the initial sample and the negative control containing only Protein G Sepharose resin (without antibody). Bars represent the mean ± SD of four independent measurements, and a statistically significant difference from the initial sample is indicated by an asterisk (*).

**Figure 4 biomedicines-13-02528-f004:**
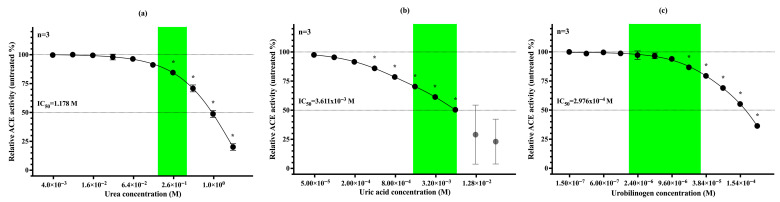
ACE activity was measured in purified urine samples (30 kDa retentate) combined with serial dilutions of urea (**a**), uric acid (**b**), and urobilinogen (**c**). Each symbol represents the mean and standard deviation of three independent measurements. IC_50_ values are indicated for each compound. The green zone indicates the physiological concentration range of each molecule in urine. Because of measurement-related limitations, the gray points serve illustrative purposes only. Significant differences from the initial value are indicated by an asterisk (*).

**Figure 5 biomedicines-13-02528-f005:**
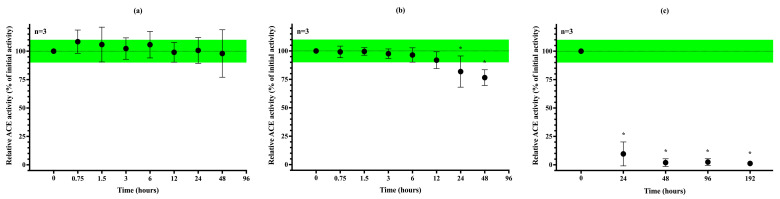
The effect of storage temperature on uACE activity was determined at +25 °C (**a**), +4 °C (**b**) and −20 °C (**c**). Each symbol represents the mean and standard deviation of three independent measurements. Significant differences from the initial value are indicated by an asterisk (*).

**Figure 6 biomedicines-13-02528-f006:**
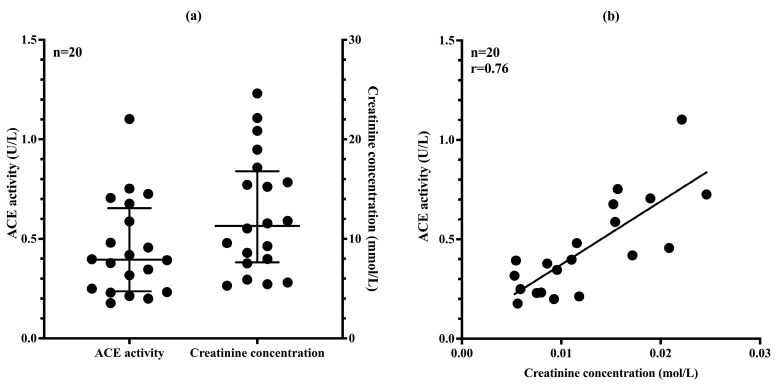
(**a**) Distribution of uACE activity and creatinine concentration in individuals not taking ACE inhibitors, shown with median values and interquartile ranges. (**b**) Correlation between uACE activity and creatinine concentration. Each point represents one individual data point.

**Figure 7 biomedicines-13-02528-f007:**
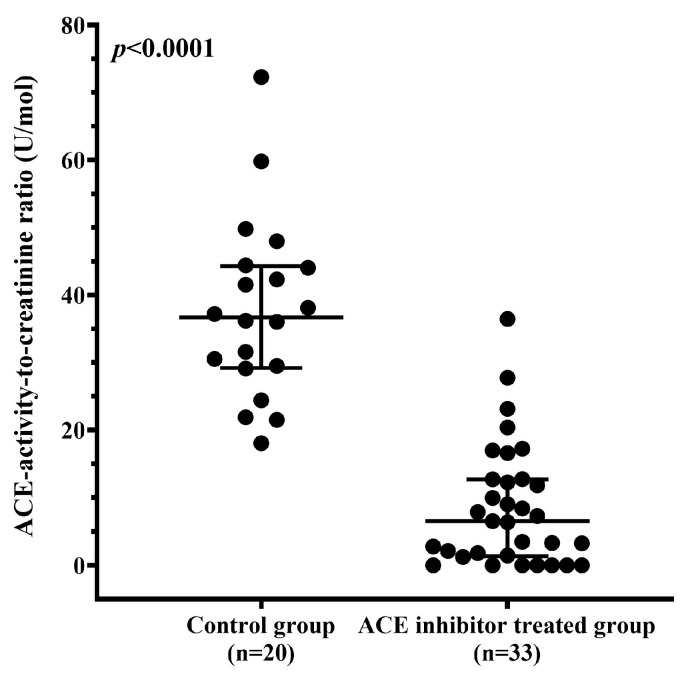
uACE activity, normalized to creatinine concentration, is shown for patients not receiving ACE inhibitors (n = 20) and for those undergoing ACE inhibitor therapy (n = 33). Each point represents an individual value. Group medians and interquartile ranges are indicated. A significant difference between groups is denoted by the *p* value.

**Table 1 biomedicines-13-02528-t001:** Demographic characteristics of the examined groups are shown. Data are presented separately for the control group and the ACEi-treated group. Continuous variables are expressed as mean ± standard deviation or median (interquartile range), as appropriate. Categorical variables are presented as counts (n).

		Control Group		Study Group	
		Women	Men	Women	Men
**n**		11	9	11	22
**Age** (years)		49.42 ± 17.32	48.04 ± 18.42	67.60 ± 7.93	70.57 ± 10.39
**BMI** (kg/m^2^)		25.64 (23.53–28.04)	28.15 ± 6.14	29.51 ± 5.61	29.80 ± 5.31
**eGFR epi** (mL/min/1.73 m^2^)		90.00 (85.00–90.00)	90.00 (86.50–90.00)	74.82 ± 19.70	81.00 (70.50–90.00)
**Cardiovascular disease** (n, (%))		5 (45%)	4 (44%)	11 (100%)	21 (95%)
**Hypertension** (n, (%))		3 (27%)	2 (22%)	9 (82%)	19 (86%)
**Diabetes mellitus** (n, (%))		3 (27%)	1 (11%)	5 (45%)	12 (55%)
**Renal failure** (n, (%))		1 (9%)	0 (0%)	1 (9%)	0 (0%)
**Smoking** (n, (%))		3 (27%)	4 (44%)	3 (27%)	4 (18%)
**ACEI drug type** (n, (%))	**perindopril**	0 (0%)	0 (0%)	10 (91%)	17 (77%)
	**ramipril**	0 (0%)	0 (0%)	0 (0%)	5 (23%)
	**lisinopril**	0 (0%)	0 (0%)	1 (9%)	0 (0%)

## Data Availability

The raw data can be obtained on request from the corresponding author.

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
