# Peer review of "Interference-Free Measurement of Urinary Angiotensin-Converting Enzyme (ACE) Activity: Diagnostic and Therapeutic Monitoring Implications"

_biomedicines, 2025, doi:10.3390/biomedicines13102528_

Round 1
Reviewer 1 Report
Comments and Suggestions for Authors
Dear
Assistant Editor
Biomedicines
I have reviewed the manuscript with ID biomedicines-3880775 with title “Interference-Free Measurement of Urinary Angiotensin Converting Enzyme (ACE) Activity: Diagnostic and Therapeutic Monitoring Implications” by the Authors:
Attila Ádám Szabó, Enikő Edit Enyedi, Tamás Bence Pintér, Ivetta Siket Mányiné, Csongor Váradi, Emese Bányai, Attila Tóth, Zoltán Papp, Miklós Fagya. where the authors proposed establish a fast and reproducible method for measuring urinary ACE activity, to identify the inhibitory compounds re-sponsible for previous assay failures, and to define practical preanalytical conditions suitable for routine laboratory implementation, The manuscript is interesting and reads fluently since they used artificial intelligence to improve the writing and structure of the manuscript. However, I have a series of questions before accepting it in your prestigious journal and that artificial intelligence has overlooked.
The authors consistently use the following phrase: Urinary angiotensin-converting-enzyme. Could you please add the "U" to the abbreviation for ACE and use it throughout the manuscript?
Use an abbreviation for cardiovascular disease (CVD) and use it throughout the manuscript when necessary.
At the beginning of the methods section, could the authors please describe the type of study conducted, e.g., analytical, comparative, or prospective study between groups: controls and patients with different types of ACE inhibitors vs. patients without inhibitors.
It's not clear to me why they used 5 subjects from their research group to measure the analyte they measured, and what did they compare that n to, since there's already a control group of 20 subjects without inhibitors.
If three patients were excluded from the control group, and the study consisted of 20, why put 23 at the beginning of the section? Please modify this.
I believe from my very personal point of view that the experimental groups are poorly designed
here we have a control group without inhibitors (n=20), then a group of patients with ace inhibitors (n=33) and a group of patients without ace inhibitors would be missing
The authors mention that patients with CVD used ACE inhibitors but they only show and mention lisonopril, in the population of 33 patients they sampled, only these patients used this inhibitor, what about the other drugs used for ACE inhibition please clarify, and if they were selected only with this inhibitor, they should include a study limitations section where this situation is clarified in order to limit their analysis method.
Since ACE is modulated by sex hormones, where inhibition is more effective in the male population, the authors should add a table with demographic data for both controls and patients, for example, age, weight, height, gender, etc.
The results must be presented in another way, since the experimental method mentions two experimental groups, the results must be presented considering both groups.
In Figure 1 a, b,c and figure 5a, figure 6 add the significant differences (p)
By using n=33 and according to how the authors present the data of median and standard deviation is not adequate and since they mention that in some 25-75 percentiles were used, then everything should be presented in 25 percentile Q1, median Q2 and 75 percentile Q3 and if we continue with this way of presenting the data since the population distribution is not parametric then why are some graphs presented with n= 3 (figure 3 and 4 a, b, c) should show the 33 data. Please clarify this situation.
Because the correlation was between patients who did not take the inhibitors (control subjects), I think this correlation doesn't make sense. The correlation should be between patients with inhibitors and creatinine concentration. Please clarify this.
In the discussion section lines 360-363 the authors mention that LC-MS/MS only quantifies plasma concentrations of ACEi concentrations, this phrase is erroneous since it is a highly sensitive method and can also be used in urine to obtain the same analyte. Please modify this.
I thank you in advance for the opportunity to review this manuscript.
Sincerely, the reviewer.
Reviewer 2 Report
Comments and Suggestions for Authors
I have carefully reviewed your manuscript “Interference-Free Measurement of Urinary ACE Activity” (Manuscript ID: biomedicines-3880775) submitted to Biomedicines.
Overall, the article is well written, clearly structured, and addresses an interesting and clinically relevant topic. The experimental work is solid, and the optimization of an interference-free assay for urinary ACE activity is an important methodological contribution. I believe that your manuscript has good potential to be published in Biomedicines, but several major issues need to be addressed before it can be accepted.
Please find below my detailed comments:
Major comments
1.- The manuscript presents a technically sound and well-documented approach to measure urinary ACE activity free from endogenous interference. While the authors emphasize its potential application as a marker of therapy adherence to ACE inhibitors, this use is likely of limited practical value in clinical practice, where blood pressure control remains the simplest and most reliable indicator of treatment efficacy. Also, it should be noted that the pharmacokinetics of ACE inhibitors differ substantially across the drug class. Hydrophilic agents such as lisinopril, enalaprilat, or captopril are excreted predominantly in urine and can indeed inhibit urinary ACE activity at therapeutic concentrations. In contrast, more lipophilic ACE inhibitors (e.g., trandolapril, fosinopril) undergo significant hepatic metabolism and biliary excretion, potentially limiting their impact on urinary ACE. Furthermore, in patients with advanced renal impairment, urinary drug excretion may be reduced, confounding the interpretation of urinary ACE activity as a surrogate of adherence. Urinary ACE activity, on the other hand, is subject to multiple confounders (dietary factors, age, renal function, urine dilution, endogenous inhibitors), which constrain its reliability as an adherence biomarker. The authors should discuss these limitations and specify that the proposed approach may not be generalizable to all ACE inhibitors or to all patient populations. However, the true strength of this work lies in the establishment of a reproducible, interference-free assay that could be exploited in other contexts—particularly as a potential biomarker of tubular injury, renal inflammation, or local RAAS-related processes. The authors are encouraged to shift the focus of the discussion toward these physiopathological and diagnostic applications, which may represent the most impactful clinical implications of their findings.
2.- The inhibitory curves presented for urea, uric acid, and urobilinogen are central to the conclusions of the study. However, the Methods section does not specify the exact amount of ACE enzyme used in these assays, nor the fitting equation applied to generate the inhibition curves. Since the apparent potency (IC₅₀) values depend critically on the enzyme concentration and on the model chosen for curve fitting (e.g., four-parameter logistic), these details are essential for reproducibility. The authors should clearly state the amount of enzyme present in the assay, the fitting equation, and the software used, and include this information either in the Methods or in the figure legends.
3.- The authors demonstrate that urea, uric acid, and urobilinogen inhibit urinary ACE activity. Given that physiological urinary concentrations of uric acid (~2–6 mM) are in the same range as the reported IC₅₀ (3.6 mM), and urea concentrations (100–500 mM) approach inhibitory levels, these molecules may indeed contribute substantially to ACE inhibition in vivo. This raises important physiological questions: in the elderly, where urea excretion is typically increased, urinary ACE activity may be more strongly suppressed. Such an age-related inhibition could represent a homeostatic mechanism to favor natriuresis and water excretion by reducing luminal bradykinin degradation. The authors should consider discussing the potential physiological implications of this inhibition, beyond the methodological aspect of assay interference.
4.- Lisinopril inhibition alone is not sufficient to unambiguously attribute the observed activity to ACE. Other zinc-dependent peptidases present in urine, such as aminopeptidase N, Xaa-Pro aminopeptidases, and potentially even matrix metalloproteinases, can be inhibited—albeit at higher concentrations—by ACE inhibitors. Therefore, the authors are encouraged to strengthen the enzyme characterization by (i) validating the specificity with multiple ACE inhibitors of different chemical classes, (ii) testing inhibitors selective for other peptidases (e.g., thiorphan for neprilysin, bestatin for aminopeptidases), and (iii) providing direct protein evidence (Western blotting, immunoprecipitation, glycosylation profiling). Without such characterization, cross-reactivity cannot be fully excluded.
5.- Origin of urinary ACE. The manuscript assumes that urinary ACE originates from shedding of the proximal tubular brush border. However, the observed correlation between urinary ACE activity and creatinine concentration is more likely to reflect dilution/concentration effects rather than proving tubular origin. In pathological contexts (e.g., sarcoidosis, nephropathies), urinary ACE may represent a mixed contribution from tubular shedding and filtered plasma-derived ACE. The authors should moderate their statements about the cellular origin of urinary ACE, and explicitly acknowledge that the present study does not provide direct biochemical evidence on this matter. Discussing the need for future studies (plasma–urine correlations, Western blotting, glycosylation profiling, exosomal ACE) would strengthen the manuscript.
6.- The authors report complete loss of urinary ACE activity after a single freeze–thaw cycle at –20 °C. This marked instability contrasts with published data on serum ACE, which is relatively stable under freezing conditions. Such a difference suggests that urinary ACE may represent a distinct, more labile form of the enzyme (e.g., a truncated/shedded or vesicular variant) rather than circulating ACE filtered into urine. The authors should explicitly discuss this discrepancy and, ideally, compare the stability of urinary versus serum ACE side by side. Further biochemical characterization (Western blot, glycosylation profiling) would be important to confirm that the labile activity measured in urine is indeed ACE and not another protease.
7.- Cohort characterization and limitations. The study includes a relatively small cohort, and the clinical characteristics of the subjects (age, sex, comorbidities, renal function, concomitant medications) are not sufficiently described. Since urinary ACE activity is influenced by renal physiology and metabolic state, this information is essential for interpreting the findings. Moreover, the manuscript would benefit from a dedicated Limitations section summarizing key constraints: uncertain cellular origin, marked instability after freezing, lack of full biochemical characterization, and the modest sample size.
Minor Comments
- Please revise typographical errors throughout the text (e.g., line 262: a closing parenthesis “)” is included without the corresponding opening “(”).
- Figure legends could be more detailed, including exact IC₅₀ values.
- Consider simplifying some long sentences in the Introduction to improve readability.
- Ensure that all abbreviations are defined at first use.
Round 2
Reviewer 1 Report
Comments and Suggestions for Authors
The authors have responded appropriately to all my suggestions, which is why the manuscript can be published in your prestigious journal.
Thank you.
Author Response
We sincerely thank the reviewer for the positive assessment and recommendation for publication.
Reviewer 2 Report
Comments and Suggestions for Authors
Manuscript ID: biomedicines-3880775
General assessment
I would like to sincerely thank the authors for the extensive and thorough revision of their manuscript. They have carefully considered all my previous comments and responded to each point with additional data, clarifications, and appropriate modifications in the text. The revised version represents a substantial improvement in both scientific quality and clarity.
The newly added experiments (including the characterization with multiple ACE inhibitors, the evaluation of cross-reactivity with other peptidase inhibitors, and the immunodepletion assays) provide convincing evidence for the specificity of the proposed assay.
Likewise, the inclusion of methodological details (enzyme activity range, curve-fitting model, and software) has enhanced reproducibility and transparency.
The authors have also revised the Discussion in line with my earlier recommendations, appropriately acknowledging the pharmacokinetic variability among ACE inhibitors, the limitations in clinical generalization, and the potential diagnostic relevance of urinary ACE as a marker of tubular injury or renal inflammation. These additions considerably strengthen the manuscript.
Specific remarks
Only minor editorial or linguistic issues remain. The manuscript would benefit from a final English language polishing by a native speaker or professional editor to improve fluency, consistency in terminology (for example, several sentences could be slightly refined for fluency, such as replacing “are the physiological inhibitors” with “act as physiological inhibitors,” or “impactful future clinical implications” with “clinically relevant implications.” These are minor stylistic improvements that would enhance readability but do not affect the scientific content), and minor stylistic details.
Final recommendation.
All major scientific concerns have been fully addressed.
I therefore recommend acceptance after minor language editing.
Comments on the Quality of English LanguageSpecific remarks
Only minor editorial or linguistic issues remain. The manuscript would benefit from a final English language polishing by a native speaker or professional editor to improve fluency, consistency in terminology (for example, several sentences could be slightly refined for fluency, such as replacing “are the physiological inhibitors” with “act as physiological inhibitors,” or “impactful future clinical implications” with “clinically relevant implications.” These are minor stylistic improvements that would enhance readability but do not affect the scientific content), and minor stylistic details.
Final recommendation.
All major scientific concerns have been fully addressed.
I therefore recommend acceptance after minor language editing.
Author Response
We are very grateful for the reviewer’s detailed and encouraging comments.
In accordance with the suggestion, we have performed a final round of language editing with the assistance of a professional scientific English editor to further improve fluency, consistency, and overall readability.
These stylistic refinements included minor rewording of several expressions (for example, “are the physiological inhibitors” → “act as physiological inhibitors,” and “impactful future clinical implications” → “clinically relevant implications”), ensuring uniform terminology throughout the text.
Importantly, these corrections were purely editorial in nature and did not alter the scientific content, data interpretation, or conclusions of the manuscript.
All such minor modifications are highlighted in yellow in the revised version for transparency.